# Neural Functional Singular Value Decomposition for Irregularly Sampled Infinite Dimensional Data

## Abstract

This work studies an extension of the singular value decomposition to infinite-dimensional spaces by considering neural networks as basis elements. In contrast to the classical finite-dimensional singular value decomposition, this approach is grid-less and can be used in cases where only irregularly sampled data are available and evaluation at arbitrary sample points is required. To the best of our knowledge, we are the first work to propose a neural rank reduction method that is capable of handling irregularly sampled data. Our approach is based on a regularized least-squares formulation that fits the neural network to the given data while enforcing normalization for each function and orthogonality between pairs of functions. Performing rank reduction for infinite-dimensional operators is particularly interesting for scientific machine learning with focus on predicting the solution of partial differential equations given some boundary condition. In this context, the learned neural basis functions form a linear and finite-dimensional approximation of the image of the solution operator. We demonstrate the efficacy of our algorithm by first learning this approximation based on given irregularly sampled data. In a second stage, we train an artificial neural network as a coefficient functional for the previously learned basis.

## 1 Introduction

Deep learning has seen remarkable success in a large variation of different tasks, ranging from computer vision (Krizhevsky et al., 2012), natural language processing (Brown et al., 2020), or playing games (Silver et al., 2016). Recently, neural networks have also found their way into the domain of scientific computing tackling challenging problems in fields such as Earth systems science (Reichstein et al., 2019) or protein folding (Jumper et al., 2021).

A central task in computational science is to solve partial differential equations (PDEs) where, historically, finite difference methods and finite element methods have been dominant (Brenner & Scott, 2008). However, in recent years, there has been significant focus on machine learning methods in order to augment the pool of available methods (Beck et al., 2023; Gonon et al., 2024). One notable approach in this area has been physics-informed neural networks (Raissi et al., 2019), which uses the residual of the PDE as part of the loss function together with some available measurements in order to train a neural network as Ansatz for the PDE. A second direction is to use a variational formulation such as in the DeepRitz method (E & Yu, 2018) to minimize an energy functional, which was further developed into methods for solving high-dimensional PDEs such as the electronic Schrödinger equation (Pfau et al., 2020; Hermann et al., 2020; Scherbela et al., 2022). In a similar vein, the Feynman-Kac formula relates the solution of a Kolmogorov PDE to the minimization of an expectation value, which was utilized by (Han et al., 2018; Beck et al., 2021) to solve very high-dimensional problems in mathematical finance such as option pricing by the Black-Scholes formula.

The most interesting topic for the present work is the field of operator learning. In this field, available simulation and measurement data are used to directly learn to predict the solution of PDEs by learning the infinite-dimensional solution operator (Li et al., 2021; Lu et al., 2021; Boullé et al., 2024; Nelsen & Stuart, 2024).

These methods all share the same underlying rationale: the data is modeled as elements in an infinite-dimensional vector space. This is sensible due to the fact, that much of the data we encounter are naturally modeled as continuous objects, functions over a domain, hence infinite-dimensional in principle. Since computers always have finite memory, practical implementations of algorithms for infinite-dimensional data require embedding the data in some finite-dimensional representation. This embedding into a latent space can be seen as using encoders and decoders (Bhattacharya et al., 2021; Lanthaler et al., 2022). Note that these finite representations of infinite-dimensional models are different from directly using a finite model, as we assume the model to have potentially infinite resolution and being sampling invariant.

Most state-of-the-art models make the assumption that the finite-dimensional training data are obtained by point sampling on regular sample points, that is, the same set of evaluation points for each function in the training data (Li et al., 2021; Lu et al., 2022). In practice, however, output data are often irregularly sampled: each training instance is observed at its own set of sensor locations due to heterogeneous meshes, moving probes, or measurement constraints.

We address this setting with a neural functional singular value decomposition (nfSVD) where we focus on finding a low-rank approximation of infinite-dimensional data with irregular point samples. We learn a continuous, orthonormal range basis directly from scattered output samples via a single regularized least-squares objective evaluated only at the observed points. To obtain a deployable surrogate, we then train a coefficient functional that maps inputs to basis coefficients. In our experiments this is instantiated with either a fully-connected feedforward network or a convolutional neural network.

Our contributions include:

- A neural singular value decomposition method for representing infinite-dimensional data. In contrast to existing methods, our method allows for multi-dimensional inputs and provides a grid-free representation.
- A novel training scheme for the neural network basis functions which guarantees orthogonality and normalization via soft penalty constraint.
- Numerical validation that our method is indeed capable of learning basis functions from both regularly and irregularly sampled data over one- and two-dimensional domains. The obtained basis functions furthermore perform well in the downstream task of operator learning.
- Simulation results on regularly sampled data show that our method achieves a performance that is comparable to the provably optimal basis of finite-dimensional proper orthogonal decomposition basis. In our simulations on irregularly sampled data, our method indeed uniformly outperforms existing state-of-the-art methods.

## 2 RELATED WORK

The main motivation for our work stems from the task of building operator surrogates. Foundational theoretical work in operator learning was done by (Chen & Chen, 1995), who proved that the universal approximation property of neural networks also holds for mappings between infinite-dimensional spaces. The goal here is to approximate some non-linear operator $\mathcal{G} \colon \mathcal{U} \to \mathcal{V}$ between Banach spaces $\mathcal{U}$ and $\mathcal{V}$. We write $v(y) = \mathcal{G}[u](y)$ to indicate that the operator $\mathcal{G}$ acts on the function $u \in \mathcal{U}$ and the resulting function $v \in \mathcal{V}$ is evaluated at the point $y \in \operatorname{dom} \mathcal{V}$.

Building on this foundation, the deep operator network (DeepONet) (Lu et al., 2021) was the first widely used operator learning architecture, which extends the shallow architectural design proposed by Chen & Chen (1995) to deep networks. To approximate the operator $\mathcal{G} \colon \mathcal{U} \to \mathcal{V}$, the approach is to consider a sequence of sampling points $\{x_i\}_{i \in \mathbb{N}}$ and the architecture

$$\hat{v}(y) = \hat{\mathcal{G}}[u; \{x_i\}_{i \in \mathbb{N}}](y) = \sum_{n \in \mathbb{N}} \Phi_n(\{u(x_i)\}_{i \in \mathbb{N}})\Psi_n(y), \qquad (2.1)$$

where $\mathbf{\Psi}$ is a basis for the infinite-dimensional output space and $\mathbf{\Phi}$ are the corresponding coefficient functionals. For practical applications, however, such a system of infinite size cannot be realized and the coefficient representations have to be truncated after finitely many elements.

In the case of regularly sampled data Lu et al. (2022) proposed the POD-DeepONet, which first approximates the covariance matrix of the training data and then uses the first $N$-dominant eigenvectors of that matrix as a low-rank approximation of the range space of the operator, that is, as the basis from 2.1. Notice that this approach, while computationally efficient, comes with the drawback of being valid only on the fixed regular sample points of the dataset.

An extension of low-rank approximation to the infinite-dimensional setting is not straightforward. Constructing the full covariance function, such as it is done in the functional principal component analysis Ramsay & Silverman (2005), is expensive and requires a lot of data (see (Tan et al., 2025)). Lee & Shin (2024) approximate the image of the operator via a non-orthogonal neural network basis. Subsequently, they perform a finite-dimensional QR-decomposition on the functions sampled on regular sample points. However, this being done as post-processing after the training bears the risk of raising generalization error as now the basis does not necessarily form a good approximation of the distribution of functional data. Note that the method as proposed in (Lee & Shin, 2024) relies on regularly sampled data.

Tan et al. (2025) developed an approach that uses elements from a reproducing kernel Hilbert space (RKHS) to perform a functional singular value decomposition for infinite-dimensional data from heterogeneous sources. For an RKHS $\mathcal{H}_K$, sampling points $\{T_{ij}|j \in [J_i]\}$, and hidden functions $X_1, \ldots, X_N \in \mathcal{H}_K$ with noisy observations $Y_{ij} = X_i(T_{ij}) + \varepsilon_{ij}$, their algorithm iteratively finds the functional singular value decomposition up to rank $R$ by finding the current leading singular component via

$$\underset{\boldsymbol{a}_1 \in \mathbb{R}^N, \phi_1 \in \mathcal{H}_K}{\arg\min} \quad \sum_{i=1}^{N} \frac{1}{J_i} \sum_{j=1}^{J_i} (Y_{ij} - a_{i1}\phi_1(T_{ij}))^2 + \nu \|\boldsymbol{a}\|^2 \cdot \|\mathcal{P}\phi_1\|_{\mathcal{H}}. \tag{2.2}$$

Here, $\nu > 0$ is a tuning parameter, and $\mathcal{P}$ is a projection operator from $\mathcal{H}_K$ onto its subspace. The subspace corresponding to $\phi_1$ is then removed from the data, and they continue sequentially for the next singular component. This approach operates in a kernel regime and is limited to one-dimensional input data.

## 3 Neural Functional Singular Value Decomposition

In this section, we present a nfSVD tailored to operator learning with irregularly sampled outputs. Unlike RKHS-based variants, we work with a neural hypothesis class $\mathbb{F}_{d,N} \subset L^2(\mathbb{D})$, a grid-free least-squares objective, and enforce $L^2(\mathbb{D})$ orthonormality via Monte-Carlo inner products. The result is a finite-rank, mesh-independent range basis suitable for multi-dimensional domains and heterogeneous sensors. In the following we now formalize the objective and its discretization.

In the conventional finite-dimensional setting, the singular value decomposition (SVD) is a method for factorizing a matrix $\boldsymbol{X} \in \mathbb{C}^{J \times M}$ into $\boldsymbol{X} = \boldsymbol{U}\boldsymbol{\Sigma}\boldsymbol{V}^T$, where $\boldsymbol{U} \in \mathbb{C}^{J \times J}$, and $\boldsymbol{V} \in \mathbb{C}^{M \times M}$ are unitary matrices and $\boldsymbol{\Sigma} \in \mathbb{R}^{J \times M}$ is a rectangular matrix with decreasing non-negative entries on the diagonal. By restricting this factorization only to the first $N < J$ columns of $\boldsymbol{U}$ and $\boldsymbol{V}$ it is known to be the best possible low-rank approximation of $\boldsymbol{X}$ by the Eckart-Young-Theorem (see (Golub & Van Loan, 2013, Theorem 2.4.8)).

In this work, we restrict to real-valued spaces and consider an extension of this to the infinite-dimensional setting. That is, instead of a matrix, we consider a linear operator $X \in \mathcal{L}(\mathbb{R}^M, \mathcal{H})$ from the $M$-dimensional Euclidean space to some separable Hilbert space $\mathcal{H}$. In this setting, we can then uniquely identify $X$ with an $M$-tuple $(x_1, \ldots, x_M) \subset \mathcal{H}$ such that $X : \mathbb{R}^M \to \mathcal{H}$ is the map $\boldsymbol{v} \mapsto \sum_{m=1}^{M} v_n x_m$. As orthonormal basis in $\mathcal{H}$ for a rank $N$ approximation of the operator $X$ with $N \leq M$ we then consider the solution of the constrained least-squares problem

$$(r_1^*, \ldots, r_N^*), \boldsymbol{A}^* \in \underset{\substack{(r_1, \ldots, r_N) \in \mathcal{H}^N, \\ \boldsymbol{A} \in \mathbb{R}^{N \times M}}}{\arg\min} \quad \sum_{m=1}^{M} \left\| x_m - \sum_{n=1}^{N} A_{n,m} r_n \right\|_{\mathcal{H}}^2 \tag{3.1}$$

$$\text{s.t.} \quad \langle r_{m_1}, r_{m_2} \rangle_{\mathcal{H}} = \mathbf{1}_{m_1 = m_2}, \, \forall m_1, m_2 \in [M].$$

For a practical implementation of this problem, we consider the Hilbert space $L^2(\mathbb{D})$ with compact domain $\mathbb{D} \subset \mathbb{R}^d$ and make the following observations:

- We do not have access to the infinite-dimensional operator $X$ but only to point samples of the data $\{x_1, \ldots, x_M\}$ and need to reformulate the minimization.

- Optimization over the full space $L^2(\mathbb{D})$ (or even its unit sphere) is impossible and we need to restrict to some parametrized hypothesis class $\mathbb{F}$.

- For non-linearly parametrized hypothesis classes, it is infeasible to enforce the orthonormality as a hard constraint and we need to resort to regularization instead.

## 3.1 DISCRETIZED FUNCTION DATA

We consider the setting, where for each function $x_m$ we have access to a certain number $T_m \in \mathbb{N}$ of sample points

$$\boldsymbol{s}^{(m)} := (s_1^{(m)}, \ldots, s_{T_m}^{(m)}) \in \mathbb{D}^{T_m} \tag{3.2}$$

and the corresponding function values

$$\boldsymbol{x}^{(m)} := (x_m(s_1^{(m)}), \ldots, x_m(s_{T_m}^{(m)})) \in \mathbb{R}^{T_m}. \tag{3.3}$$

Based on the sample points, we then also define for the basis functions $r_1, \ldots, r_N \in L^2(\mathbb{D})$ the representation

$$\boldsymbol{R}^{(m)} := (r_n(s_t^{(m)}))_{t \in [T_m], n \in [N]} \in \mathbb{R}^{T_m \times N}. \tag{3.4}$$

We consider two major cases: regularly sampled data and irregularly sampled data. Here, regular sampling means that the sample points stay constant over different data functions, that is, $T_{m_1} = T_{m_2}$ and $\boldsymbol{s}^{(m_1)} = \boldsymbol{s}^{(m_2)}$ for all $m_1, m_2 \in [M]$. Note, however, that this does not necessarily mean that the sample points lie on a regular grid. Contrary to that, irregular sampling means, that the sample points vary for different data functions.

Based on the data discretization, the loss function will also be discretized to

$$\left\| x_m - \sum_{n=1}^N A_{n,m} r_n \right\|_{L^2(\mathbb{D})}^2 \approx \frac{|\mathbb{D}|}{T_m} \left\| \boldsymbol{x}^{(m)} - \boldsymbol{R}^{(m)} A_{:,m} \right\|_2^2, \tag{3.5}$$

which is a statistically unbiased estimator, given that the sample points are i.i.d. uniform in $\mathbb{D}$ or on a regular grid.

Similarly to that, we also approximate the inner product. To do so, we build the union over all available sample points and the corresponding matrix representation of the basis functions

$$\bar{\boldsymbol{s}} := (\boldsymbol{s}^{(1)}, \ldots, \boldsymbol{s}^{(M)}) \in \mathbb{D}^T, \quad \text{and} \quad \bar{\boldsymbol{R}} := (r_n(\bar{s}_t))_{t \in [T], n \in [N]}, \tag{3.6}$$

respectively, with $T = \sum_{m=1}^M T_m$. The resulting approximation is then

$$\langle r_{n_1}, r_{n_2} \rangle_{L^2(\mathbb{D})} \approx \frac{|\mathbb{D}|}{T} \bar{R}_{:,n_2}^T \bar{R}_{:,n_1}, \tag{3.7}$$

which is again unbiased if the sample points are i.i.d. uniform in $\mathbb{D}$ or on a regular grid.

## 3.2 HYPOTHESIS CLASS AND ORTHONORMAL REGULARIZATION

As hypothesis class for the basis functions, we consider artificial neural networks (ANNs) with $d$ input neurons and $N$ output neurons and a fixed hidden architecture, which we denote by $\mathbb{F}_{d,N}$. The hidden architecture is not important for the current discussion and will be specified later in the experiments in Section 4. Note here, that $f \in \mathbb{F}_{d,N}$ is not necessarily a tuple of $N$ functions with $d$-dimensional input but rather a functions $f : \mathbb{R}^d \to \mathbb{R}^N$. However, it is clear that $\mathbb{F}_{d,N} \subset L^2(\mathbb{D})^N$.

Unlike kernel methods, which are naturally formulated in a Hilbert space, ANNs do not have a canonic way of forcing the orthogonality of individual functions $r_n$ in the parameter space. Enforcing the orthogonality a posteriori bears the risk of worsening performance due to the non-linear nature of neural network parametrization. Thus, directly enforcing the constraint $\langle r_{n_1}, r_{n_2} \rangle_{L^2(\mathbb{D})} =$

$\mathbf{1}_{n_1=n_2}$ is practically infeasible. Instead, we reformulate the constraint into regularization terms of the form $(\langle r_{n_1}, r_{n_2}\rangle_{L^2(\mathbb{D})} - \mathbf{1}_{n_1=n_2})^2$, which are weighted by regularization parameters $\tau_N$ for normalization (i.e., $n_1 = n_2$) and $\tau_O$ for orthogonalization (i.e., $n_1 \neq n_2$). Notice that we also enforce normality of the neural basis functions to avoid individual functions collapsing to zero or individual modes becoming too dominant. In this way, we can ensure that the basis has a more equitable weight across the spectrum. Here, directly enforcing the constraint would be possible by rescaling the last layer. However, since gradient methods for neural network training such as ADAM (Kingma & Ba, 2017) are based on moment estimation, this kind of additional projection step would worsen optimization results.

Overall, the final learning problem is given by

$$(r_1^*, \ldots, r_N^*), \boldsymbol{A}^* \in \underset{\substack{(r_1,\ldots,r_N)\in\mathbb{F}_{d,N} \\ \boldsymbol{A}\in\mathbb{R}^{N\times M}}}{\arg\min} \quad \sum_{m=1}^{M} \frac{1}{T_m} \left\| \boldsymbol{x}^{(m)} - \boldsymbol{R}^{(m)} A_{:,m} \right\|_2^2 \tag{3.8}$$

$$+ \tau_N \frac{|\mathbb{D}|}{T^2 N} \sum_{n=1}^{N} \left( \|\bar{R}_{:,n}\|_2^2 - 1 \right)^2 + \tau_O \frac{|\mathbb{D}|}{T^2 N(N-1)} \sum_{\substack{n_1=1 \\ n_1\neq n_2}}^{N} (\bar{R}_{:,n_2}^T \bar{R}_{:,n_1})^2.$$

## 4 EXPERIMENTS

For the experimental analysis of our algorithm we look at the downstream task of learning infinite-dimensional operators (see Section 2). That is, we assume that we are given the initial condition or some parameters (the input) of the PDE and aim to predict the solution (the output). In this setting, the training data are composed of point samples taken from $M$ pairs of input- and output-functions $k\colon \mathbb{D}_I \to \mathbb{R}^{d_I}$ and $x\colon \mathbb{D}_O \to \mathbb{R}^{d_O}$, respectively. Here, $\mathbb{D}_I, \mathbb{D}_O, d_I, d_O$ represent the input and output domain and the dimensions of the input and output functions, respectively. In all our experiments we will assume that $d_I = d_O = 1$ and $\mathbb{D}_I = \mathbb{D}_O = [0,1]^d =: \mathbb{D}$. For each input function $k_m$ with $m \in [M]$ we are then given a set of $S_m \in \mathbb{N}$ sample points $\boldsymbol{y}^{(m)} \in \mathbb{D}^{S_m}$ with the corresponding function samples $\boldsymbol{k}^{(m)} \in \mathbb{R}^{S_m}$. Analogously, for the output functions $x_m$, we are given $T_m \in \mathbb{N}$ sample points $\boldsymbol{s}^{(m)} \in \mathbb{D}^{T_m}$ with the corresponding function samples $\boldsymbol{x}^{(m)} \in \mathbb{R}^{T_m}$.

For the operator learning task, we then assume that the input data is regularly sampled, that is, $\boldsymbol{y}^{(m_1)} = \boldsymbol{y}^{(m_2)} =: \boldsymbol{y}$ with $S_{m_1} = S_{m_2} =: S$, while we consider both cases of regularly and irregularly sampling for the output functions.

In all our experiments, we consider a set of 1000 pairs of input output functions for training and a set of 200 pairs for testing. As parameters, we set the regularization parameters to $\tau_O = 10^{-3}$, $\tau_N = 10^{-5}$, an initial learning rate (for both steps) of $10^{-3}$, and a weight decay regularization of magnitude $10^{-4}$. The learning rate is reduced with a geometric decay starting after 20000 epochs for the basis functions and with an inverse time decay for the coefficient functionals, both at a rate of $10^{-4}$. Differences to this standard setting will be mentioned when necessary.

Our main error metric will be the relative $L^2(\mathbb{D})$-error which is averaged over five independent initializations of the neural networks. Note, that the random sampling of our data is kept constant over these individual runs such that the statistics of the $L^2(\mathbb{D})$-error solely reflect the variations in the model and not the variations in the dataset.

### 4.1 OPERATOR LEARNING VIA NFSVD

In order to apply our algorithm to the operator learning task, we identify the collection of output functions $(x_1, \ldots, x_M)$ with the operator $X\colon \mathbb{R}^M \to L^2(\mathbb{D})$. We then use our algorithm (3.8) in order to find a neural network $\boldsymbol{\Psi}^*$ which provides a low-rank approximation for the span of $X$. For this learned basis and the corresponding coefficient matrix $\boldsymbol{A}^*$, we train another network $\boldsymbol{\Phi}$, with the aim that each $\Phi_n$ is the coefficient functional for the $n$-th basis function. These functionals are trained to fit the data given by the samples of the input functions and the corresponding coefficients in the matrix $\boldsymbol{A}^*$ of (3.8) in a least-squares sense, thus conforming to the operator learning formalism of (2.1).

A similar two-stage training method to learn an orthonormal basis of neural networks was proposed in form of the QR-DeepONet (Lee & Shin, 2024), which aims to reduce the complexity of the training procedure for DeepONets. In this approach, the first step is to train a neural network $\mathbf{\Psi} \in \mathbb{F}_{d,N}$ on the output data using a least-squares objective. In a second stage, the network is transformed into a matrix representation by evaluating on the sampling points. By means of a QR-decomposition of the matrix representation the approach obtains a reweighting matrix $\mathbf{R}$, whose inverse is applied to the neural network $\mathbf{\Psi}$ such that the basis functions are orthonormal. Note, that in the original description of Lee & Shin (2024), the approach is limited to regularly sampled data. We extend this formulation to irregularly sampled data, by considering the matrix representation $\bar{\mathbf{R}}$ on the union of all sample points as we have done in our approach in (3.6).

Furthermore, we compare to the predecessors of QR-DeepONet, namely the Vanilla DeepONet Lu et al. (2021) and the POD-DeepONet Lu et al. (2022) where the network $\mathbf{\Psi}$ is replaced by a matrix $\mathbf{P} \in \mathbb{R}^{T \times N}$. In order to ensure a fair comparison between the different models, we use the same hyperparameters for training whenever possible. For example, our approach and QR-DeepONet use the same optimizer, initial learning rate and learning rate scheduler for the training of the basis functions in $\mathbf{\Psi}$. Furthermore, the output functions are rescaled by $\sqrt{N}$ according to the second order analysis of (Lu et al., 2022).

## 4.2 Problem Setup

For the experiments we resort to some of the commonly used test settings of the operator learning literature. Namely, the one dimensional viscous Burgers equation (Li et al., 2021; Lu et al., 2022), Darcy flow on a rectangular domain with continuous permeability field (Lu et al., 2022), and Darcy flow on a rectangular domain with piecewise constant permeability field (Li et al., 2021; Lu et al., 2022). In the following we proceed with a detailed description of our datasets and method of random sampling. [1]

### 4.2.1 Burgers Data

We consider the one dimensional viscous Burgers equation

$$\partial_t u(x,t) + \partial_x(u^2(x,t)/2) = \nu \partial_{xx} u(x,t), \qquad x \in (0,1),\, t \in (0,1] \qquad (4.1)$$
$$u(x,0) = u_0(x), \qquad x \in (0,1) \qquad (4.2)$$

with periodic boundary conditions and viscosity $\nu = 0.1$. This dataset originates from (Li et al., 2021), where initial condition $u_0$ was sampled from $\mathcal{N}(0, 625(-\Delta + 25I)^{-2})$ and the data was generated with a spatial resolution of $2^{13}$ points. The task of operator learning is then to consider the input $k = u_0$ and predict the solution $x = u(\cdot, 1)$ at time $t = 1$.

For our simulations on regularly sampled data, we subsample all data to a spatial resolution of 128 points with uniform spacing. For irregularly sampled output data, we perform a random subsampling of the output data, which means, a subset of 128 unique sample points is picked uniformly from the original $2^{13}$ points. Examples of input-output pairs for both regular and irregular sampling are shown in Fig. 1.

In order to enforce the periodic boundary conditions, we adopt the approach of (Lu et al., 2022) to transform the spatial variable

$$s \mapsto (\cos(2\pi s), \sin(2\pi s), \cos(4\pi s), \sin(4\pi s))$$

at the input of $\Psi$. The architecture of our networks for this case is that both $\mathbf{\Psi}$ and $\mathbf{\Phi}$ have 3 hidden layers with 128 neurons and $\tanh$ activation function, and we consider $N = 32$ basis functions. We train the basis functions for 50000 epochs and the coefficient functionals for 1000000 epochs.

### 4.2.2 Darcy Continuous Permeability Field

An example for a two-dimensional problem for operator learning is given by the Darcy-Flow problem on a rectangular domain with a log-normal random field as permeability field, constant one Dirichlet boundary condition on the left, zero Dirichlet boundary condition on the right, and zero

---

[1] A link to the source code will be made available upon publication

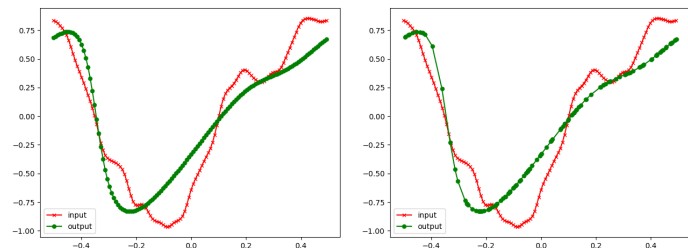

Figure 1: Examples of input-output pairs of the Burgers dataset for regularly sampled output (left) and irregularly sampled output (right).

Table 1: Results for Burgers equation with regular sampling and irregular sampling

|  | regular sampling | | irregular sampling | |
|---|---|---|---|---|
|  | rel. $L^2$ | proj. rel. $L^2$ | rel. $L^2$ | proj. rel. $L^2$ |
| nfSVD | $\mathbf{1.39 \pm 0.14}\,\%$ | $0.19\,\%$ | $\mathbf{1.46 \pm 0.09}\,\%$ | $0.18\,\%$ |
| POD (Lu et al., 2022) | $\mathbf{1.39 \pm 0.04}\,\%$ | $0.21\,\%$ | N/A | N/A |
| QR (Lee & Shin, 2024) | $1.58 \pm 0.23\,\%$ | $\mathbf{0.17}\,\%$ | $1.62 \pm 0.19\,\%$ | $\mathbf{0.15}\,\%$ |
| VAN (Lu et al., 2021) | $1.70 \pm 0.04\,\%$ | $0.36 \pm 0.01\,\%$ | $1.80 \pm 0.05\,\%$ | $0.41 \pm 0.01\,\%$ |

Neumann boundary condition on the top and bottom boundary of the domain. That is, we consider the problem

$$-\nabla \cdot (k(\boldsymbol{s})\Delta x(\boldsymbol{s})) = 0, \qquad\qquad \boldsymbol{s} \in (0,1)^2, \qquad\qquad (4.3)$$

$$x(\boldsymbol{s}) = 1 - s_1, \qquad\qquad s_1 \in \{0,1\},\ s_2 \in (0,1), \qquad\qquad (4.4)$$

$$\partial_n x(\boldsymbol{s}) = 0, \qquad\qquad s_1 \in (0,1),\ s_2 \in \{0,1\} \qquad\qquad (4.5)$$

with the permeability field $k(s) = \exp(f(s))$ where $f$ is a two dimensional Gaussian process with covariance kernel $\exp\left(\|s\|_2^2/0.5^2\right)$. For this dataset we could not reuse the data of (Lu et al., 2022) as we needed a higher resolution available for random subsampling. Instead, we use the FEniCS finite element simulation tool (Alnæs et al., 2015) to simulate the PDE with a resolution on a grid with $401 \times 401$ points. For the regularly sampled case, we then subsample this to a resolution of $21 \times 21$ points, resulting in 441 points. For the irregularly sampled case, we then subsampled from the two-dimensional data by randomly selecting 441 unique sample points. An example is given in Fig. 2.

The architecture of our networks for this case is that $\boldsymbol{\Psi}$ has 3 hidden layers of width 40 with $\tanh$ activation function, $\boldsymbol{\Phi}$ is a convolutional neural network with $\tanh$ activation function, and we consider $N = 10$ basis functions. We train the basis functions for 50000 epochs and the coefficient functionals for 100000 epochs.

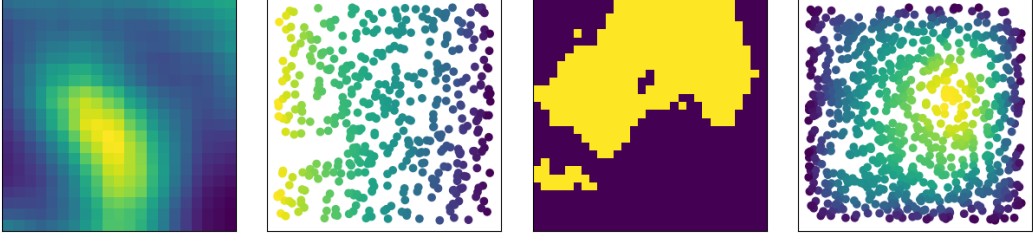

Figure 2: Example of input-output pair for Darcy flow on a rectangular domain with regularly sampled input and irregularly sampled output. The plots on the left show one example of for the continuous permeability field and the plots on the right show one example of a piecewise constant permeability field.

Table 2: Results for Darcy flow with continuous permeability field with regular sampling and irregular sampling

|  | regular sampling | | irregular sampling | |
| --- | --- | --- | --- | --- |
|  | rel. $L^2$ | proj. rel. $L^2$ | rel. $L^2$ | proj. rel. $L^2$ |
| nfSVD | $\mathbf{1.12 \pm 0.02}\,\%$ | $0.79\,\%$ | $\mathbf{1.48 \pm 0.03}\,\%$ | $0.98 \pm 0.03\,\%$ |
| POD (Lu et al., 2022) | $1.13 \pm 0.02\,\%$ | $\mathbf{0.78}\,\%$ | N/A | N/A |
| QR (Lee & Shin, 2024) | $1.14 \pm 0.04\,\%$ | $\mathbf{0.78}\,\%$ | $1.51 \pm 0.03\,\%$ | $\mathbf{0.92}\,\%$ |
| VAN (Lu et al., 2021) | $1.15 \pm 0.02\,\%$ | $0.79 \pm 0.01\,\%$ | $1.56 \pm 0.12\,\%$ | $0.97 \pm 0.04\,\%$ |

Table 3: Results for Darcy flow with piecewise constant permeability field with regular sampling and irregular sampling

|  | regular sampling | | irregular sampling | |
| --- | --- | --- | --- | --- |
|  | rel. $L^2$ | proj. rel. $L^2$ | rel. $L^2$ | proj. rel. $L^2$ |
| nfSVD | $2.27 \pm 0.01\,\%$ | $1.06\,\%$ | $\mathbf{2.79 \pm 0.17}\,\%$ | $1.38 \pm 0.21\,\%$ |
| POD (Lu et al., 2022) | $2.26 \pm 0.01\,\%$ | $\mathbf{0.92}\,\%$ | N/A | N/A |
| QR (Lee & Shin, 2024) | $\mathbf{2.25 \pm 0.01}\,\%$ | $0.93\,\%$ | $15.12 \pm 25.04\,\%$ | $\mathbf{0.98 \pm 0.06}\,\%$ |
| VAN (Lu et al., 2021) | $2.61 \pm 0.02\,\%$ | $1.15 \pm 0.01\,\%$ | $2.98 \pm 0.03\,\%$ | $1.05\,\%$ |

### 4.2.3 DARCY PIECEWISE CONSTANT PERMEABILITY FIELD

We also tested our method for another instantiation of the Darcy flow problem in two dimensions. Here, we chose homogeneous Dirichlet boundary condition of zero and a piecewise constant permeability field given by

$$k(s) = \begin{cases} 12, & \text{if } T > 0 \\ 3, & \text{otherwise} \end{cases}$$

where $T$ was sampled by a Gaussian random field with a kernel of $\mathcal{N}(0, (-\Delta + 9I)^{-2})$ and zero Neumann boundary condition. This problem poses an interesting challenge as the spectrum of the covariance operator is decaying very slowly. This then also accounts for the generally high number of basis functions needed for a good approximation. The original data has a resolution of $421 \times 421$, which we subsampled to a grid $29 \times 29$ points in the regular case and 841 randomly selected but unique points in the irregular case. We used the raw data from the repository of (Lu et al., 2022).

The architecture of our networks for this case is that $\mathbf{\Psi}$ has 4 hidden layers with 256 neurons and $\tanh$ activation function, $\mathbf{\Phi}$ is a convolutional neural network with $\tanh$ activation function, and we consider $N = 115$ basis functions. The hyperparameters in this experiments deviate from the standard setting in the sense that $\tau_O = 10^{-2}$, $\tau_N = 10^{-4}$ and the learning rate decay for the training of the basis functions has a rate of $5 \cdot 10^{-5}$. We train the basis functions for 50000 epochs and the coefficient functionals for 100000 epochs.

### 4.3 DISCUSSION

The resulting relative $L^2(\mathbb{D})$ error for both sampling methods is presented in Tables 1 to 3 alongside the resulting relative $L^2(\mathbb{D})$ error of the projection of the output functions onto the learned basis $\mathbf{\Psi}$ for nfSVD, QR-DeepONet, and DeepONet and $P$ for the POD-DeepONet. We observe that the relative $L^2(\mathbb{D})$ error of our model consistently shows the best results in the irregularly sampled output case, while achieving the performance of the POD-DeepONet for the regularly sampled case. It is important to note that while the proper orthogonal decomposition used as basis in this architecture is provably optimal for rank reduction, both the QR-DeepONet and nfSVD network achieve lower projection error in our experiments on the Burgers data. The reason for that lies that we take the relative $L^2(\mathbb{D})$ projection error and both networks are trained on normalized function samples, whereas the POD-DeepONet architecture is build upon unnormalized data.

An interesting question arises from the projection error of the modified QR-DeepONet being generally lower than from the architecture based on nfSVD. This fact is however not surprising as the

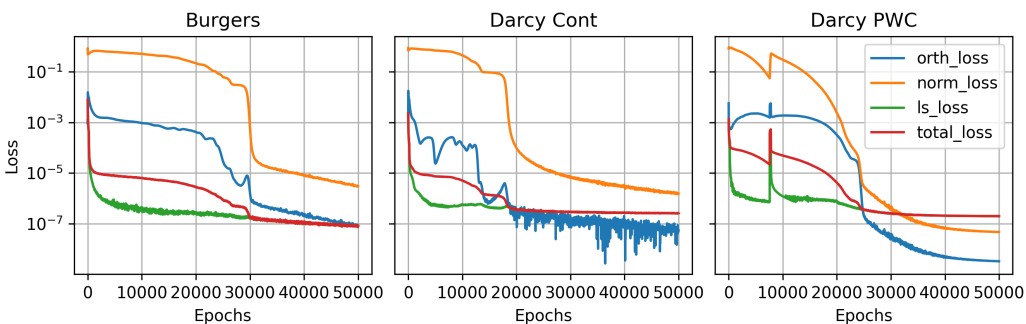

Figure 3: Convergence of nfSVD for irregularly sampled data on Burgers data (left), Darcy flow with continuous permeability field (middle) and Darcy flow with piecewise constant permeability field (right).

QR-DeepONet does not possess regularization terms, leaving data fit to be the sole objective. The generalization loss, on the contrary, shows better results for the regularized nfSVD approach, highlighting the possible advantage of such constraints. This is also visible in the last problem (see Table 3), where the unregularized approach failed to converge in some of the runs resulting in very high error.

Using a soft penalty constraint for orthonormality opens up the question to which degree we can actually achieve this in our neural network basis. As can be seen from Fig. 3, the normalization loss experiences a step-wise behavior which culminates in a one-time sharp drop at which also the orthogonality loss shows a significant drop. This behavior is explained by the fact that throughout our simulations we set the orthogonality parameter $\tau_O$ to be two orders of magnitude larger than the normality parameter $\tau_N$. With this setting, the network $\mathbf{\Psi}$ is initially close to the zero function and whenever one of its outputs opens up a new linearly independent direction in $L^2(\mathbb{D})$, this output converges quickly to unit norm, resulting in the step of the normalization loss. The single significant drop then indicates that the last output converged to unit norm. As a consequence of our choice of $\tau_O$ and $\tau_N$ the normalization loss is generally higher than the orthogonality loss, however, we see that we indeed achieve the initial goal of learning orthonormal functions up to numerical inaccuracy (see Appendix B).

## 5 CONCLUSION

In this work, we introduced the neural functional singular value decomposition and applied it to learning operator surrogates. We have shown empirically that this method produces stable, accurate orthonormal approximations of the range space of different relevant PDE solution operators. The nfSVD consistently shows superior results on irregularly sampled data, while achieving results identical to the provably optimal row-rank approximation on shared grids.

There are several avenues of research going from here. A straightforward extension could be to utilize the nfSVD for encodings of the input space, either on its own or in combination with an output space encoding. Sequential training of neural basis functions could bring the benefit of finding the dominant modes first and lead to greater similarity of the learn basis with a proper orthogonal decomposition in the discrete space. Solving the least-squares problem for the basis coefficients could lead to a potentially even faster convergence at the price of a much more costly backpropagation. Additionally, exploring natural gradient descent or subspace methods for the basis coefficients could prove viable. An extension in a new direction outside the scope of this paper is given by non-linear low rank approximations. Instead of approximating the range space with a linear approximation given by an $N$-dimensional vector space, one might use non-linear elements parameterizing an $N$-dimensional manifold potentially leading to better or more efficient approximation of the structure of, for example, a non-linear operator.

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

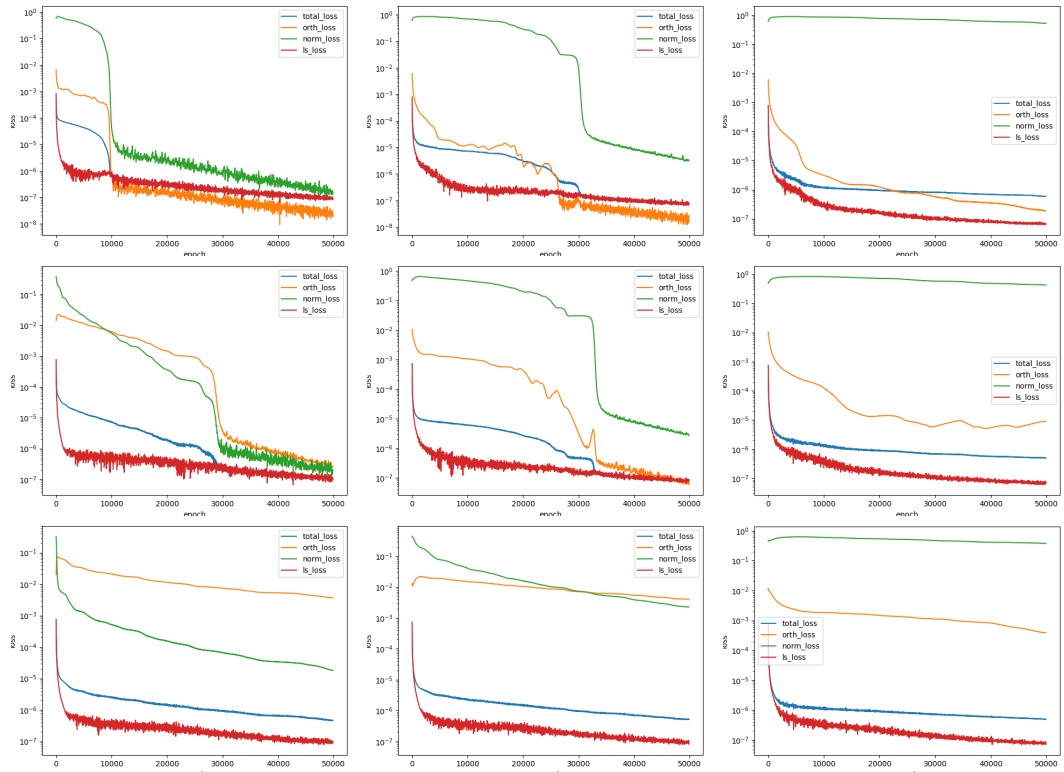

Figure 4: Loss curves for different choices of the regularization parameters $\tau_O$ and $\tau_N$. From top to bottom we vary $\tau_O \in \{10^{-2}, 10^{-3}, 10^{-4}\}$ and from left to right we vary $\tau_N \in \{10^{-4}, 10^{-5}, 10^{-6}\}$.

## A  BURGERS REGULARIZATION PARAMETERS

In this section we want to provide a short ablation study on the influence of differing regularization parameters for the problem of solving the Burgers equation 4.2.1. We tested the parameters $\tau_O$ and $\tau_N$ at different orders of magnitude, among which we show all combinations for the orthogonality parameter $\tau_O \in \{10^{-2}, 10^{-3}, 10^{-4}\}$ and the normality parameter $\tau_N \in \{10^{-4}, 10^{-5}, 10^{-6}\}$ in Fig. 4. It is clearly visible that the smallest choice $\tau_O = 10^{-4}$ results in non-convergence of the orthogonalization loss within 50000 epochs, while the smallest choice $\tau_N = 10^{-6}$ results in non-convergence of the normalization loss. We observe that higher regularization parameters result in faster convergence, however, the combination $\tau_O = 10^{-3}$ and $\tau_N = 10^{-5}$ results in the lowest possible least squares loss. This justifies our specific choice of parameters in our experiments.

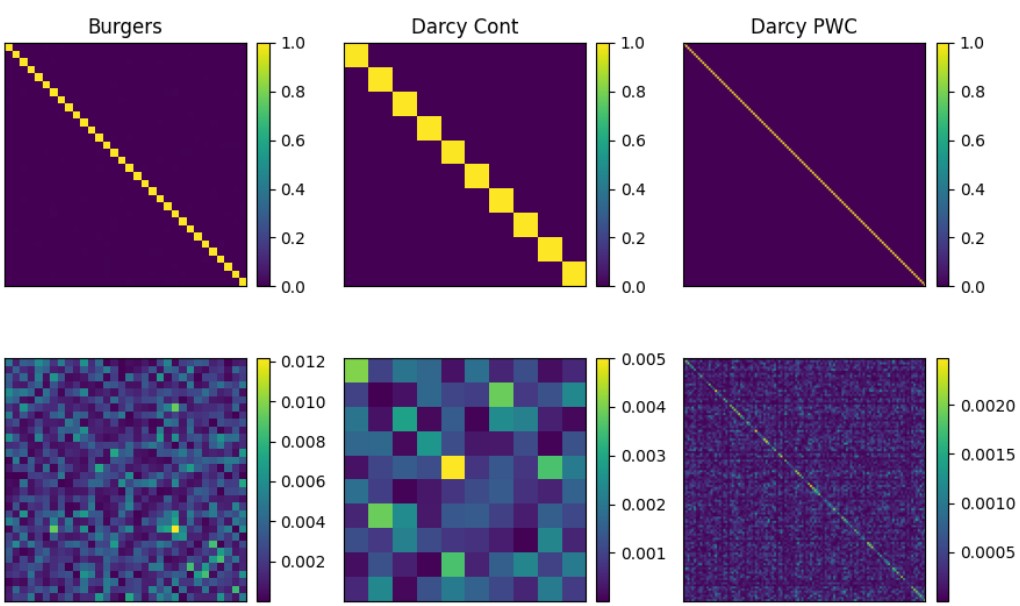

Figure 5: Correlation matrices (upper row) for basis functions and the corresponding error matrices (bottom row). The matrices are obtained from irregularly sampled data on the three datasets considered in our experiments.

## B  ORTHOGONALITY VISUALIZATION

In Fig. 5 we visualize the orthogonality and error in orthogonality in order to demonstrate the quality of our solutions. We do this by computing the correlation matrix $\frac{|\mathbb{D}|}{T}\bar{R}^T\bar{R}$ with $|\mathbb{D}| = 1$ and the absolute deviation of this matrix from the identity matrix. Orthogonality is an important property of a low-rank approximation, as it guarantees efficiency in the number of modes used. As functions are orthogonal exactly then, when their inner product is zero, a pure penalty of correlation tends to give outputs that are very small in norm. It is well known in machine learning, that optimization tends to work better on normalized functions, as such, building operator surrogates with a coefficient functional would prove much more unstable and error-prone.

