# OpenReview forum: "Neural Functional Singular Value Decomposition for Irregularly Sampled Infinite Dimensional Data"
_ICLR.cc/2026/Conference — ICLR 2026 Conference Withdrawn Submission_

### Official Review · Reviewer_5qvv · 2025-10-21

**Soundness:** 2
**Presentation:** 2
**Contribution:** 2
**Rating:** 2
**Confidence:** 4

**Summary:**

This work considers an extension of the singular value decomposition to infinite dimensions to compute low-rank approximation to operators between function spaces. The aim of the method is to perform rank-reduction using irregularly sampled data, by fitting a low-rank neural network with an appropriate regularization term to weakly enforce orthonormality. The authors provide numerical experiments on several benchmark problems to illustrate the performance of the method, and compare it to other low-rank approximation methods.

**Strengths:**

- The method is able to handle arbitrary sampling of the input data, which is a significant advantage over alternative neural operator techniques that require structured grids.
- The numerical experiments are competitive with other approaches on regularly sampled grids.

**Weaknesses:**

- The novelty of the method is limited, as it primarily consists of training a finite rank neural network with a regularization term to weakly enforce orthonormality. It is not clear from section 3.2, why the regularization term is necessary, instead of just having a low-rank approximation and then orthonormalizing the functions after training. Moreover, the constraint is only weakly enforced during training and the inner products are approximated using Monte Carlo integration, which may lead to inaccuracies, and suppose that the sample points are uniformly sampled. I also disagree with the claim in p.2 that the training scheme guarantees orthonormality, as it is only weakly enforced through a regularization term. A more sensible approach would be to use a quadrature rule adapted to the sampling of the data (e.g. a finite element representation, or a Fourier discretization for Fourier neural operators).
- The idea of using neural operators with a low-rank structure has been explored in previous works (see Low-rank neural operators in Kovachki et al. 2023), and the variations of DeepONet mentioned by the authors. Here, one would typically want a nonlinear approximation of the PDE solution operators, as opposed to a finite-rank linear one as solution operators associated with PDEs are typically nonlinear, and not of numerical low-rank when the PDE is linear. It is therefore not clear whether the contribution of this work is significant and would bring substantial improvements over existing methods.
- The numerical experiments are limited in scope as the authors only compared against variations of DeepONet in two simple problems (while FNO-based architectures are more popular), and show small improvements over vanilla DeepONet.

**Questions:**

- How does one select the target rank $N$ in practice? Is there a way to adaptively increase it during training if the desired accuracy is not met?
- How does the method depend on the choice of the regularization parameter $\tau_O$?

---

### Official Review · Reviewer_7RKz · 2025-10-31

**Soundness:** 2
**Presentation:** 1
**Contribution:** 2
**Rating:** 2
**Confidence:** 5

**Summary:**

The paper introduces a continuous extension of the Singular Value Decomposition (SVD) algorithm in which neural networks serve as basis functions. Orthonormality of the learned bases is encouraged through a soft constraint incorporated into the loss function. The proposed method is evaluated on two benchmark partial differential equations: the Burgers equation and the Darcy flow problem.

**Strengths:**

The mathematical framework presented is reasonable and has promising potential for model reduction in settings with irregularly sampled functions. The orthogonalization term is formulated correctly, and the experiment show promise.

**Weaknesses:**

- **Related work**. The most significant weakness concerns the discussion of related literature. In particular, paper [1] appears to propose a very similar approach (including an orthogonalization term), yet it is not cited or discussed. Although I was only able to locate a withdrawn OpenReview submission and not an arXiv version, the contribution still seems relevant and should be acknowledged, along with a clear explanation of the differences between the two methods. Additionally, other approaches addressing orthogonalization have been developed in [2, 3]; these works should also be cited and compared against.
- **Experimental evaluation**. The experimental section currently feels limited. Only two test cases are presented, and the results seem to indicate that QR could serve as a competitive alternative to the proposed method. It remains unclear in which situations the proposed approach provides a clear advantage. Furthermore, there is no discussion of hyperparameter tuning, which makes it difficult to assess whether the chosen configurations are optimal — both for the proposed model and for the baselines.
- **Clarity and notation**. The paper can be challenging to follow at times, particularly due to inconsistent notation. For instance, the symbol $x$
is sometimes used to denote sampling points (e.g., Equation 2.1) and elsewhere to represent a function (e.g., Equation 3.3). Ensuring consistent notation throughout would greatly improve readability.

***References***
[1] Demo, Nicola, Dario Coscia, and Gianluigi Rozza. "A space-continuous implementation of Proper Orthogonal Decomposition by means of Neural Networks, 2024.
[2] Larrazabal, Agostina J., et al. "Orthogonal ensemble networks for biomedical image segmentation." International Conference on Medical Image Computing and Computer-Assisted Intervention. Cham: Springer International Publishing, 2021.
[3] Mashhadi, Peyman Sheikholharam, Sławomir Nowaczyk, and Sepideh Pashami. "Parallel orthogonal deep neural network." Neural Networks 140 (2021): 167-183.

**Questions:**

- How robust are the results to different number of modes?
- How does your method scale with the number of modes both in parameters size, and training and inference time?

---

### Official Review · Reviewer_MBZP · 2025-10-31

**Soundness:** 2
**Presentation:** 1
**Contribution:** 2
**Rating:** 2
**Confidence:** 4

**Summary:**

The paper proposes a *neural functional singular value decomposition* (nfSVD) framework for analyzing a set of functions (typically presumed to be solutions of a PDE). Given a set of functions $\\{x_1(\cdot), ..., x_M(\cdot)\\}$, where $x_m\\colon \\mathcal{X}\\to \\mathbb{R}$, the goal is to find basis vectors $\\mathbf{a}_m\in\mathbb{R}^N$ for $m=1,\\ldots,M$ and basis functions $r_i\\colon\\mathcal{X}\\to\\mathbb{R}$ for $i=1,\ldots,N$ such that
$$x_m\approx \mathbf{a}_m^{\intercal} \mathbf{r}\_{1:N}$$
for all $m\in[M]$, where $\mathbf{r}\_{1:N}$ denote the stack of the basis functions.
The authors propose to minimize the approximation error in the Frobenius norm
$$
\\sum\_{m=1}\^M \\|x\_m - \\mathbf{a}\_m\^{\\intercal} \\mathbf{r}\_{1:N} \\|\_{\\mathcal{H}}\^2
$$
while enforcing approximate orthonormality between the neural basis functions through soft penalty terms,
to learn the best neural-network basis functions $\mathbf{r}\_{1:N}$ and best matrix $A=[\\mathbf{a}_1^\intercal,\ldots,\\mathbf{a}_M^\intercal]^\intercal$.
By Eckart-Young-Mirsky theorem, this will capture the best rank-$N$ approximation, which is the rank-$N$ truncated SVD, of the $M\times |\mathcal{X}$ dimensional object $(x_1,\ldots,x_M)$.
The authors emphasize the benefit that the method can handle when each function is "irregularly discretized" and claim that this is the first method that possess such ability.
The paper further discusses applications to PDE operator surrogates and presents empirical results on the Burgers and Darcy flow benchmarks.

**Strengths:**

The paper proposes a variant of the functional SVD (fSVD) framework to incorporate neural networks and irregularly sampled data.

**Weaknesses:**

- First of all, the paper is very hard to follow conceptually. The main problem is that the authors never define precisely *what* object is being decomposed. The target operator only appears halfway through Section 3, and the discussion around DeepONet (Eq. 2.1) makes it sound as if the authors are trying to perform an SVD of a nonlinear operator, which does not make mathematical sense. The problem setup needs to be made explicit from the beginning: what is the operator, what space does it act on, and what data is given.

- The treatment of prior work is also superficial. The paper essentially builds upon the formulation of the **functional SVD** (Tan et al., 2025) but replaces the RKHS basis with neural networks. If there is a conceptual difference, it is not articulated. The authors should first explain the functional SVD properly before presenting their neural variant, instead of the informal remark in Section 2. Consider putting Related Work somewhere else for better flow.

- The notation and exposition are heavier than necessary and obscure the main ideas.

- Several equations are sloppy or even incorrect. For example, line 152 writes $(x\_1,\\ldots,x\_M)\\subset\\mathcal{H}$, which does not seem correct. These kinds of inconsistencies make it difficult to understand what problem is actually being solved.

- On the novelty claim: `we are the first to handle irregularly sampled data with a neural rank-reduction method` is wrong. Earlier works such as SpIN [A] and NeuralSVD [B] already use neural singular/eigen functions to decompose differential operators using irregularly sampled data (in ODE/PDE settings). Moreover, [B] also uses the low-rank approximation as a variational principle to characterize the top-k singular subspaces. [Appendix B.1.5, B] has an overview on this line of works, including, e.g., [C]. These earlier works should be acknowledged.

- The term *operator learning* is used loosely throughout the paper. Section 4.1 (`Operator Learning via nfSVD`) is especially misleading, as the operators to be decomposed in the paper is in a very specific form. This is misleading as "operator learning" in the DeepONet context means a completely different concept.

- The orthogonality regularization is also questionable. The paper adds soft penalties for orthogonality and normalization, but a low-rank approximation can be learned without such constraints (see [B]), so this choice needs justification.

- Last but not least, the experiments show only marginal improvements, and many design choices (hyperparameters) are unexplained.

```
[A] Pfau, David, et al. "Spectral inference networks: Unifying deep and spectral learning." ICLR 2018.
[B] Ryu, J. Jon, et al. "Operator SVD with neural networks via nested low-rank approximation." ICML 2024.
[C] Ben-Shaul, Ido, et al. "Deep learning solution of the eigenvalue problem for differential operators." Neural Computation 35.6 (2023): 1100-1134.
```

**Questions:**

- `We train the basis functions for 50000 epochs and the coefficient functionals for 100000 epochs.` Why are there two numbers of epochs? What is the training procedure?
---

#### **Suggestions**

The paper would benefit from a complete reorganization. Consider the following suggestions:

- Define the problem setup and the target operator clearly at the start.
- Introduce the functional SVD formally before presenting the neural version.
- Then explain the neural functional SVD, focusing on how this parameterization allows irregular sampling.
- Correct the novelty claim and discuss prior neural spectral-decomposition work (SpIN, NeuralSVD, etc.).
- Simplify notation, use consistent terminology, and justify the need for orthogonality regularization.
- Include quantitative ablations (e.g., effect of orthogonality penalties).

---

#### **Final Remarks**

The idea of handling irregular sampling in functional SVD is interesting, but the paper is conceptually muddled and overstated. The problem setup is unclear, the theory is weak, and the novelty over existing neural spectral methods is minimal. The empirical improvements are small and not well-explained. I recommend a major revision before this work can be considered ready for publication.

---

### Official Review · Reviewer_7k65 · 2025-11-01

**Soundness:** 3
**Presentation:** 3
**Contribution:** 3
**Rating:** 4
**Confidence:** 4

**Summary:**

This paper proposes a neural functional SVD (nfSVD) for learning low-rank operator approximations from irregularly sampled data. The paper tackles a significant and practical challenge in operator learning: handling irregularly sampled output data. This is a common scenario in scientific applications that is not well addressed by many existing methods. The experiments are comprehensive and show clear advantages in the intended setting. However, the lack of theoretical analysis and limited comparison with recent neural operator methods weaken its overall strength.

**Strengths:**

The core idea is innovative and addresses a clear gap in the literature. Introducing some mathematical structures to learning-based algorithm is nice.

**Weaknesses:**

1. The comparisons are limited to the DeepONet family (Vanilla, POD, QR). A comparison with other neural operator architectures, such as extended Fourier Neural Operators (FNO) for irregular domain is missing.
2. There is no theoretical analysis, even if preliminary (e.g., a universal approximation theorem for the nfSVD representation or a convergence analysis under simplified assumptions).
3. The paper does not discuss the computational cost, training time, or scalability of nfSVD compared to the baselines. The two-stage training process is likely more expensive than single-stage end-to-end models, and this overhead should be addressed.

**Questions:**

see the Weaknesses part.

---

### Note · Authors · 2025-11-19

I have read and agree with the venue's withdrawal policy on behalf of myself and my co-authors.